# The Pandora’s Box of Frailty Assessments: Which Is the Best for Clinical Purposes in TAVI Patients? A Critical Review

**DOI:** 10.3390/jcm10194506

**Published:** 2021-09-29

**Authors:** Omar Baritello, Annett Salzwedel, Simon H. Sündermann, Josef Niebauer, Heinz Völler

**Affiliations:** 1Department of Rehabilitation Medicine, Faculty of Health Sciences Brandenburg, University of Potsdam, 14469 Brandenburg, Germany; omar.baritello@fgw-brandenburg.de; 2Research Group Molecular and Clinical Life Science of Metabolic Diseases, Faculty of Health Sciences Brandenburg, University of Potsdam, 14476 Potsdam, Germany; annett.salzwedel@fgw-brandenburg.de; 3Department of Cardiovascular Surgery, Charité—Universitätsmedizin Berlin, 10117 Berlin, Germany; simon.suendermann@charite.de; 4Department of Cardiothoracic and Vascular Surgery, German Heart Center Berlin, 13353 Berlin, Germany; 5DZHK (German Center for Cardiovascular Research), Partner Site Berlin, 13353 Berlin, Germany; 6University Institute of Sports Medicine, Prevention and Rehabilitation and Research Institute of Molecular Sports Medicine and Rehabilitation, Paracelsus Medical University, A-5020 Salzburg, Austria; j.niebauer@salk.at

**Keywords:** frailty tool, TAVI, older patients, elderly, cardiology, mortality

## Abstract

Frailty assessment is recommended before elective transcatheter aortic valve implantation (TAVI) to determine post-interventional prognosis. Several studies have investigated frailty in TAVI-patients using numerous assessments; however, it remains unclear which is the most appropriate tool for clinical practice. Therefore, we evaluate which frailty assessment is mainly used and meaningful for ≤30-day and ≥1-year prognosis in TAVI patients. Randomized controlled or observational studies (prospective/retrospective) investigating all-cause mortality in older (≥70 years) TAVI patients were identified (PubMed; May 2020). In total, 79 studies investigating frailty with 49 different assessments were included. As single markers of frailty, mostly gait speed (23 studies) and serum albumin (16 studies) were used. Higher risk of 1-year mortality was predicted by slower gait speed (highest Hazard Ratios (HR): 14.71; 95% confidence interval (CI) 6.50–33.30) and lower serum albumin level (highest HR: 3.12; 95% CI 1.80–5.42). Composite indices (five items; seven studies) were associated with 30-day (highest Odds Ratio (OR): 15.30; 95% CI 2.71–86.10) and 1-year mortality (highest OR: 2.75; 95% CI 1.55–4.87). In conclusion, single markers of frailty, in particular gait speed, were widely used to predict 1-year mortality. Composite indices were appropriate, as well as a comprehensive assessment of frailty.

## 1. Introduction

The proportion of older people (>65 years) is predicted to constantly increase in Europe and reach around 30% of the total population by 2060 [1]. These significant changes to society’s structure must be taken into account by the healthcare system. In fact, it is known that aging is associated with several geriatric syndromes, such as cognition impairments, malnutrition, sarcopenia and frailty [2]. The clinical condition of frailty especially has gained international attention in the last decade [3].

Frailty is a complex clinical condition [4], where physiological deteriorations related to the aging process are accentuated and vulnerability to stressors increases [5,6]. According to the definition of physical frailty, malnutrition, unintentional weight loss, decreased strength and endurance as well as impaired physiological functions characterize the frail population [4,7]. Frail patients have a higher risk of death and compromised independence in activities of daily living [8]. Therefore, in older patients with cardiovascular disease (CVD) undergoing elective transcatheter aortic valve implantation (TAVI), frailty should be evaluated [9]. 

Older TAVI patients are characterized by decreased physiological functionality (e.g., reduced gait speed), multiple comorbidities and malnutrition, with the prevalence of frailty found to be up to 63% in this population [5,6,10]. Pre-interventional assessment of frailty should be considered alongside common risk scores (e.g., STS score, EuroSCORE) for a better appreciation of the post-interventional prognosis [11]. In the last decade, several investigations have assessed the association of pre-interventional frailty status as an independent predictor of post-interventional mortality. However, due to the lack of consensus on the definition of frailty (e.g., physical frailty, multidimensional) and its measurement (e.g., single markers, composite indices, categorical scales, claims data), the evidence is contradictory [10,12,13,14] and identification of the most appropriate assessment for clinical practice remains unclear. 

Therefore, the aim of our review was to evaluate which frailty assessment is the most commonly used and meaningful for the prediction of short-term, intermediate-term and long-term all-cause mortality in older TAVI patients. 

## 2. Materials and Methods

The electronic database PubMed (May 2020) was systematically searched. Study inclusion criteria were the following: randomized controlled trials (RCTs) or observational studies (prospective and retrospective, respectively) without time restrictions that quantitatively assessed frailty pre-interventionally in older (≥70 years) patients who underwent elective TAVI. The outcome of interest was the association of frailty with all-cause mortality. Identification, screening, eligibility and inclusion processes were performed according to PRISMA guidelines. The included articles were reviewed by two independent researchers. The overall risk of bias in each study was evaluated using the Quality in Prognosis Studies (QUIPS) [15] and the Newcastle-Ottawa Scale (NOS) [16] tools. According to the QUIPS, we classified studies with four or five low-risk domains as having a low risk of bias overall, studies with two or more high-risk domains as having a high risk for bias overall and the remaining studies as showing a moderate risk of bias overall. For the NOS, the sum of positive adjudications was calculated. Studies were then divided into 3 categories: single markers of frailty, multidimensional frailty scales and composite indices. The single markers category comprised studies that assessed only one component of frailty, like physical function, activities of daily living (ADL), blood parameters, body composition, nutritional status and cognition. The multidimensional frailty scales category comprised studies that rated frailty using categorical scales based on clinician judgment. The third category of composite indices encompassed studies that assessed frailty based on an index resulting from the measurement of several frailty components. Furthermore, studies were stratified into short-term (≤30-day), intermediate (>30-day to <1-year) and long-term (≥1-year) mortality. 

## 3. Results

In total, *n* = 79 studies were included for full-text review (Figure 1) and *n* = 49 different frailty assessments were identified. All studies were of observational design and most of the findings referred to short- and long-term mortality. The majority of the studies investigated frailty using single markers of frailty, such as physical functionality (e.g., gait speed, handgrip strength), blood parameters (serum albumin), impairments in activities of daily living (ADL), nutritional status/risk (Mini-Nutritional Assessment), cognitive impairments (Mini-Mental State Examination) and body composition (Psoas muscle area index), as shown in Table 1. Several studies used composite indices or multidimensional scales based on clinical judgment (Table 2), and only a few studies defined frailty status based on claimed medical records and comorbidities (Appendix A). 

In Appendix A displays the characteristics of all included studies and the assessment of bias. This is followed by a summary of the results of the most commonly used assessments (Table 3). 

### 3.1. Single Markers of Frailty 

Gait speed (Table 1) was the most commonly used single marker of frailty [17,18,19,20,21,22,23,24,25,26,27,28,29,30,31,32,33,34,35,36,37,38,39], with short-term mortality assessed in *n* = 4 prospective [17,20,32,33] (9474 patients) and *n* = 6 retrospective [18,19,21,31,34,35] (62,880 patients) investigations presenting conflicting information. Kiani et al. [19] showed in their retrospective analysis (36,242 participants) that slower gait speed or being unable to walk were predictive of death rate at 30 days. Similarly, Afilalo et al. [17] and Alfredsson et al. [20] found a higher risk of mortality in participants for slow walking speed. However, six authors [21,31,32,33,34,35] stated that slow gait speed was not associated with mortality. Long-term mortality was assessed in *n* = 7 prospective [22,24,25,26,28,37,39] (2360 patients) and *n* = 9 retrospective [18,19,21,23,27,29,30,34,38] (44,222 patients) investigations and was associated with results of 5-mWT [18,19,22,23,24,25,26,27,28,29,30].

Serum albumin level was the second most commonly used single marker of frailty and was associated with short-term mortality in *n* = 6 retrospective studies [18,19,21,34,35,40]. Long-term mortality was investigated in *n* = 4 prospective [22,39,43,46] (3845 patients) and *n* = 10 retrospective [18,19,21,23,34,38,41,42,44,45] (44,062 patients) studies and the majority of them found an inverse association between levels of serum albumin and risk of death at 1 year [23,34]. 

Activities of Daily Living (ADL) were evaluated in *n* = 3 prospective [47,51,52] (825 patients) and *n* = 4 retrospective [18,21,48,49] (9822 patients) studies with conflicting results. Two retrospective investigations [48,49], of 6339 and 2624 patients reported up to a twofold higher risk of mortality for a Katz [76] score < 6. However, this was not confirmed in three other studies [21,51,52] with smaller sample sizes. Long-term mortality was assessed in *n* = 9 prospective [24,25,26,39,47,50,51,52,53] (2531 patients) and *n* = 6 retrospective [18,21,23,38,48,49] (4069 patients) investigations and contradictory information was found. 

Handgrip strength was assessed in *n* = 2 prospective [32,51] (1428 patients) and *n* = 3 retrospective [18,21,34] (4546 patients) studies, whereas decreased strength values (BMI normalized/sex stratified) were not predictive of short-term mortality. Long-term mortality was assessed in *n* = 4 prospective [24,25,26,39] (561 patients) and *n* = 5 retrospective [18,21,23,34,38] (5132 patients) investigations showing conflicting information. Five authors [18,23,24,25,26] found lower handgrip strength to be an independent predictor of mortality. In contrast, Steinvil et al. [21] and Hermiller et al. [34] showed that lower handgrip values were not predictive of relative mortality 1 year after intervention in 498 and in a larger cohort of 3687 participants, respectively.

Instrumental Activities of Daily Living (IADL) [77] were assessed in *n* = 9 prospective studies [9,12,14,22,33,50,51,54,55] (1972 patients). Most of these studies found that any impairment in the IADL scale was not predictive of short-term [9,33,51], intermediate [14,54] or long-term [9,12,22,55] mortality. Only two prospective investigations [50,51] of relatively small cohorts (116 and 213 patients) identified a higher risk of intermediate mortality for patients that presented at least two impairments.

The Mini-Mental State Examination questionnaire [78] was used in *n* = 3 prospective [9,33,51] studies with relatively small cohorts (89 to 213 participants) to predict short-term mortality. Only Storteky et al. [9] showed a higher relative risk of mortality with a decrease in cognitive capacity. Long-term mortality was investigated in *n* = 5 prospective [9,12,22,51,55] (1498 patients) and *n* = 1 retrospective [40] (1542 patients) studies, with most of the studies reporting no association with MMSE scores. Only *n* = 2 studies [9,55] with small sample sizes found that pre-interventional lower MMSE results were associated with mortality. Intermediate mortality was investigated in two prospective [14,54] studies (119 and 150 patients) that showed divergent results.

Psoas muscle area index (PMAi) was investigated in only *n* = 2 prospective studies [56,57] with conflicting evidence for short-term mortality. Kofler et al. [56] found that higher PMAi values were predictive of a lower relative risk of mortality, whereas van Mourik et al. [57] stated no association between PMAi and death rates. Long-term mortality was investigated in *n* = 5 prospective [25,56,57,58,60] (2172 patients) and *n* = 3 retrospective [44,45,59] (2515 patients) studies. The majority of these studies showed that lower PMAi seems to be associated with a negative post-interventional prognosis. 

The Timed Up-and-Go test (TUG) was used in *n* = 7 prospective studies [9,12,14,33,51,54,55] with cohorts ranging from 89 [33] to 344 [12] patients. Storteky et al. [9] found linear (each 5 sec increase) and dichotomized (≥20 versus <20 s) results on the TUG, predictive of short- and long-term mortality in 100 patients. Similarly, Eichler et al. [12] found that a longer time needed to complete the TUG (≥10 to <20 versus <10 s) was predictive of up to a fivefold relative risk of mortality in 344 patients. However, Goudzwaard et al. [51] concluded that only linear values were predictive of 1-year mortality, but not over the short-term, and only when results were dichotomized (>20 versus ≤20 s). 

### 3.2. Multidimensional Frailty Scales

The Clinical Frailty Scale (CFS) [79] was used in *n* = 9 studies [13,32,37,40,61,62,63,64,65]. Short-term mortality was considered in *n* = 2 prospective [13,32] (2235 patients) and *n* = 1 retrospective [40] (1542 patients) studies, with contradictory results. While two investigations [32,40] showed higher risk of 1-year mortality in patients with higher CFS scores, Afilalo et al. [13] demonstrated no association between CFS ≥ 5 and relative risk of mortality. Long-term mortality was investigated in *n* = 4 prospective [13,37,62,64] (3387 patients) and *n* = 4 retrospective [40,61,63,65] (2269 patients) studies, presenting divergent results.

The Canadian Study of Health Aging scale (CSHA) [79] was not predictive for short-term mortality [48,49,52]. Long-term mortality was assessed in *n* = 3 prospective [24,25,26] (402 patients) and *n* = 2 retrospective [48,52] (2936 patients) studies, that showed conflicting evidence.

### 3.3. Composite Indices

A 4-item index was used in *n* = 4 prospective [13,38,39,66] (1498 patients) and *n* = 3 retrospective [23,67,68] (650 patients) analyses. Afilalo et al. [13] found that dichotomized results predicted a more than twofold higher relative risk of short-term mortality. Similarly, Chauhan et al. [23] showed, for scores ≥ 3, a threefold higher risk of 1-year mortality in a cohort of 342 participants.

A 5-item index was used in *n* = 5 prospective [13,69,70,71,72] (2653 patients) and *n* = 1 retrospective [21] (498 patients) studies. Rogers et al. [69] found in a cohort of 544 patients that index values ≥ 3 corresponded to a fivefold increase in the relative risk of mortality at 30 days and up to a twofold increase at 1 year post-TAVI. Further, two authors [13,21] stated that higher scores on the index were predictive of long-term mortality. 

The Elderly Frailty Toolset (EFT) was used in *n* = 4 prospective [13,37,74,75] (2676 patients) and *n* = 1 retrospective [73] (723 patients) studies. Three authors [13,73,74] found higher scores of the EFT predictive of short-term mortality. Four investigations [13,37,73,74] found higher scores of the EFT to be predictive of 1-year mortality, showing up to a threefold higher risk of death in frail patients [13]. 

The Bern scale was used in *n* = 5 investigations [9,12,13,14,55], all of a prospective design (1913 patients). Afilalo et al. [13] found scores ≥ 3 to be predictive of up to a threefold higher relative risk of mortality over the short-term. Similarly, Storteky et al. [9] found a threefold higher risk of death in a cohort of 100 patients per each point increase of the index and for dichotomized results.

## 4. Discussion

Besides a distinct heterogeneity of frailty definitions and assessments used across the included studies, the main finding of our systematic review is that single markers of frailty are the most frequently used tools to identify frailty in older TAVI patients, especially parameters of physical functionality (gait speed) and malnutrition (serum albumin level). Patients with a slower walk speed (≥6 s) measured by the 5-mWT showed a higher relative risk of 1-year mortality. Pre-interventional lower levels (≤3.5 g/dL) of serum albumin were associated with a worse long-term prognosis as well. Further, higher scores of composite indices (5-item, EFT, Bern scales) were associated with higher mortality risk over the short and long term after intervention. 

A great number of studies (*n* = 23) investigated gait speed with the 5-mWT and found an association with higher 1-year mortality in older TAVI patients. Slower gait speed is largely used to evaluate impaired health conditions, especially among the elderly [80], and to assess the prevalence of frailty among the general population in a primary care setting [81]. Moreover, it is advocated as a reliable and valid tool to investigate frailty in patients with cardiovascular disease [6]. Gait speed is the result of neuromuscular control, cardiopulmonary condition, physical activity level, patient health status and sensorial pattern interactions, properly expressing the general physical functionality of older patients [82]. According to the physical frailty concept (physical phenotype), slower gait speed is considered a valid clinical indicator of frailty and sarcopenia [7]. Generally, TAVI patients who need ≥6 s (≥0.85 m/s) to perform the 5-mWT are considered frail. However, several investigations advocate that patients’ stratification into “slower/slow/normal” might be more appropriate, with the slowest walkers showing the highest risk of long-term mortality [20,39].

Besides physical functionality, malnutrition was mainly assessed as an indicator of frailty considering pre-interventional levels of serum albumin. In the majority of studies, a level of ≤3.5 g/dL was associated with a worse long-term prognosis. Albumin is a protein synthetized in the liver, and a serum concentration under 3.5 g/dL is considered representative of malnutrition [83]. Alteration of serum concentration levels is influenced by vascular injury, renal injury or various cytokine levels, and has been associated with increased risk of mortality in patients with cardiovascular disease [46]. In TAVI patients, serum albumin levels are routinely assessed pre-interventionally [46] and are often used for retrospective analysis. Especially in older and very old patients, malnutrition is a common problem correlated to several factors, such as appetite reduction, physiological changes, altered hormonal responses, mental impairments and chewing or swallowing problems [7]. An impaired nutritional intake contributes to a worsened physical functionality as well as to the process of muscle wasting and sarcopenia, which are closely related to frailty syndrome [84]. 

Composite indices were developed as measurement tools with the intent to assess several patient components involved in the clinical condition of frailty. Depending on the index, patients with three or more impaired components are considered frail. In our review, several studies found that the assessment of frailty according to composite indices was predictive of 30-day and 1-year mortality in older TAVI patients, in particular the EFT and the 5-item index. Our findings are in line with the recommendation made by the International Conference on Frailty and Sarcopenia Research (ICFSR) [85]. Initially, Fried et al. [86] recommended the evaluation of frailty according to the frail phenotype, an index based on the assessment of slowness, weakness, low physical activity, exhaustion and shrinking. These components were measured according to gait speed, handgrip strength, calculation of kilocalories expended per week, self-reported exhaustion (questionnaire) and >5 kg unintentional weight loss. Several authors revised Fried’s index and used alternative measurements tools. Thus, analysis of serum albumin levels as an expression of malnutrition was used to investigate the shrinking component [39]. Further, other indices (EFT, Bern scale) [9,13] were applied to the physical and nutritional component assessments of cognition (MMSE) and disability scales (BADL, IADL). 

In this review, we provide a detailed overview of the frailty assessments most commonly utilized in clinical practice, divided by category and their ability to predict mortality risk. However, due to the remarkable heterogeneity of the included studies, meta-analyses of the study results were not indicated. Therefore, this review does not provide pooled effect sizes for the frailty assessments investigated, potentially limiting information for the clinician. Heterogeneity is notable especially in regard to the methodology of frailty measurements, study designs and statistical analyses. For example, seven different cut-off values were identified for gait speed. This variety of cut-off values might have influenced the estimation of patients considered as frail across the studies. For composite indices, different measurement approaches (different tools used to assess the same domain; diverse domain cut-off values) were evident across studies. Additionally, the lack of a general agreement on the definition of frailty [6] and the diversity of TAVI populations included in the studies contributed to the wide range and heterogeneity of the reviewed outcomes.

First of all, there are two fundamentally different approaches to conceptualizing frailty. On the one hand, Fried et al. (2001) describe frailty based on the phenotype, focusing on physical components (e.g., unintentional weight loss, muscle weakness, slow walking speed, low physical activity and exhaustion) [86]. One the other hand, the multidimensional concept developed by Rockwood and Mitnitski (2001) comprises psychological and social components, multi-morbidity, disability in addition to the physical impairments [79,87]. Against this background, several authors purposed a large number of frailty assessments operationalizing these frailty concepts. Recently (2019), the International Conference on Frailty and Sarcopenia Research (ICFSR) recommended the Fried method for the clinical assessment of frailty [85]. However, the majority of studies in this review are conducted and published before 2019.

Although frailty is a multidimensional syndrome among older adults characterized by a marked vulnerability and diminished capability to recover from stressors, a considerable number of single markers of frailty were used in the included studies. From the statistical point of view, in multivariable analyses of mortality, single parameters often show equivalent or stronger predictive effects than composite measures [12,88,89]. Therefore, the use of single frailty markers—also due to their good practicability—in clinical studies seems justified. Given the high complexity of frailty syndrome and frailty definition, for clinical decision-making, composite indices are more appropriate for the characterization of a patient. The use of single parameters, despite being acceptable for clinical studies, is not recommended in this context.

For patients scheduled for TAVI and classified as frail, the effect of prehabilitation programs on morbidity and mortality is currently being investigated [90]. Taking into account the fact that aortic stenosis as well as frailty usually develop over years, the monitoring of malnutrition and physical function seems to be just as important in preventive cardiology as the assessment of aortic stenosis.

## 5. Conclusions

Frailty was most commonly assessed using single markers of frailty, especially based on measurements of gait speed (5-mWT) and pre-interventional levels of serum albumin. Slow gait speed (≥6 s) and a level of serum albumin ≤ 3.5 g/dL were predictive of a higher risk of 1-year mortality in older TAVI patients. Composite indices (5 items), considered as a comprehensive assessment of frailty, were associated with short- and long-term mortality. However, considerable heterogeneity was observed among the studies, and the methodology of each frailty assessment should be carefully considered. 

## Figures and Tables

**Figure 1 jcm-10-04506-f001:**
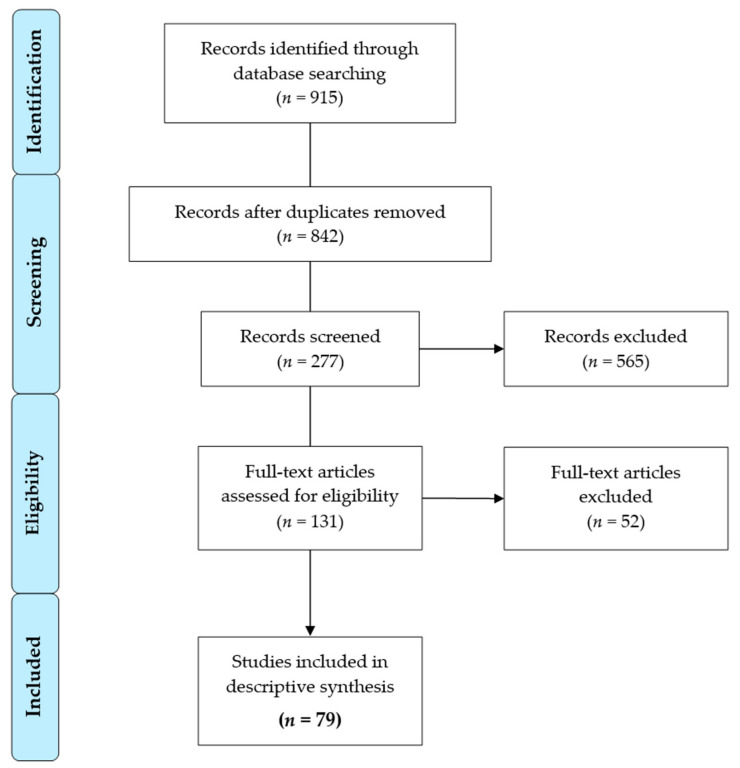
Flow diagram displaying study screening, eligibility and inclusion.

**Table 1 jcm-10-04506-t001:** Summary of the most-used single markers of frailty found to be predictive or not predictive (number of studies and (total patients)) of mortality risk over the short term (≤30 days), intermediate term (>30 days to <1 year) and long term (≥1 year).

Number of Studies and Assessment Type (Frailty Cut-Off)		Time Frame of Mortality Assesment	Predictive of Mortality Risk	
			Predictive	Not Predictive
**Single Markers of Frailty**				
23	Gait Speed (5-mWT; ≥6 s/6 MWT; unable/slow walker)	Short-term Intermediate Long-term	4 (44,773) [17,18,19,20] - 11 (41,376) [19,21,22,23,24,25,26,27,28,29,30]	6 (27,581) [21,31,32,33,34,35] 1 (232) [36] 5 (5206) [18,34,37,38,39]
16	Serum albumin (≤3.5 g/dL)	Short-term Long-term	5 (42,330) [18,19,21,34,40] 13 (48,844) [19,21,22,23,34,35,39,40,41,42,43,44,45,46]	1 (431) [35] 2 (605) [18,38]
15	ADL (Katz-index; ≥1 impaired activity)	Short-term Intermediate Long-term	4 (9624) [18,47,48,49] 2 (2740) [48,50] 6 (1405) [18,23,24,25,26,47]	3 (1023) [21,51,52] - 8 (5195) [21,38,39,48,50,51,52,53]
11	Handgrip strength (BMI normalized/sex stratified)	Short-term Long-term	- 5 (1105) [18,23,24,25,26]	5 (5974) [18,21,32,34,51] 4 (4588) [21,34,38,39]
9	IADL (Lawton-index; ≥1 impaired activity)	Short-term Intermediate Long-term	- 1 (116) [50] 2 (329) [50,51]	3 (402) [9,33,51] 2 (269) [14,54] 4 (1285) [9,12,22,55]
9	MMSE (score ≥ 27/30)	Short-term Intermediate Long-term	1 (100) [9] 1 (119) [14] 2 (430) [9,55]	2 (302) [33,51] 1 (150) [54] 4 (2610) [12,22,40,51]
9	PMAi (CT-scan; tertile BSA normalized/sex stratified)	Short-term Intermediate Long-term	1 (1076) [56] 1 (232) [36] 6 (2804) [25,45,56,57,58,59]	1 (583) [57] - 2 (1883) [44,60]
7	TUG (≥20 s)	Short-term Intermediate Long-term	1 (100) [9] 1 (119) [14] 3 (774) [9,12,55]	2 (302) [33,51] 1 (150) [54] 1 (213) [51]

5-mWT: 5-m Walk Test; 6MWT: 6-m Walk Test; ADL: Activities of Daily Living; IADL: Instrumental-ADL; MMSE: Mini-Mental State Examination; PMAi: Psoas Muscle Area index; BSA: Body Surface Area; TUG: Timed Up-and-Go test.

**Table 2 jcm-10-04506-t002:** Summary of the most-used multidimensional frailty scales and composite indices found to be predictive or not predictive (number of studies and (total patients)) of mortality risk over the short term (≤30 days), intermediate term (>30 days to <1 year) and long term (≥1 year).

Number of Studies and Assessment Type (Frailty Cut-Off)		Time Frame of Mortality Assessment	Predictive of Mortality Risk	
			Predictive	Not Predictive
**Multidimensional Frailty Scales**				
9	CFS (score ≥ 5/9)	Short-term Long-term	2 (2757) [32,40] 4 (3557) [13,40,61,62]	1 (1020) [13] 4 (2099) [37,63,64,65]
6	CSHA (score ≥ 5/7)	Short-term Intermediate Long-term	1 (6339) [49] - 4 (3026) [24,25,26,48]	2 (2936) [48,52] 1 (2624) [48] 1 (312) [52]
**Composite indices**				
7	4 items (score ≥ 3/4; gait speed, serum albumin, handgrip strength, ADL)	Short-term Long-term	3 (1254) [13,39,66] 6 (1957) [13,23,38,39,66,67]	2 (533) [23,68] -
6	5 items (score ≥ 3/5; gait speed, handgrip strength, exhaustion, low physical activity, unintentional weight loss)	Short-term Intermediate Long-term	3 (2062) [13,21,69] 1 (137) [70] 5 (3014) [13,21,69,71,72]	- - -
5	EFT (score ≥ 3/5; chair rise test, MMSE, serum albumin, hemoglobin)	Short-term Long-term	3 (2502) [13,73,74] 4 (3257) [13,37,73,74]	- 1 (142) [75]
5	Bern scale (score ≥ 3/7; MMSE, MNA, TUG, BADL, IADL, pre-clinical mobility disability)	Short-term Intermediate Long-term	2 (1120) [9,13] 1 (119) [14] 3 (1450) [9,13,55]	- - 1 (344) [12]

CFS: Clinical Frailty Scale; CSHA: Canadian Study of Health Aging; ADL: Activities of Daily Living; IADL: Instrumental-ADL; MMSE: Mini-Mental State Examination; BADL: Basic-ADL; TUG: Timed Up-and-Go test; MNA: Mini-Nutritional Assessment; EFT: Elderly Frailty Toolset.

**Table 3 jcm-10-04506-t003:** Summary of the outcomes (multivariate analysis) of the most-used assessments for all-cause mortality.

Assessment Type and Authors	Frailty Cut-Off (n.s.: Not Specified)	Mortality		
		Short-Term	Intermediate	Long-Term
**Gait speed: 5-mWT or 6MWT**				
Afilalo et al., 2010 [17]	≥6 s	OR 3.17 (95% CI 1.17–8.59)		
Chauhan et al., 2016 [23]				HR 2.62 (95% CI 1.25–5.52)
Dziewierz et al., 2017 [26]		HR 14.71 (95% CI 6.50–33.30)		
Forcillo et al., 2017 [18]		predictive		OR 0.45; *p* = 0.06
Hermiller et al., 2016 [34]		not predictive		HR 1.42 (95% CI 1.06–1.91)
Kiani et al., 2020 [19]		dichotomous ≤ 0.83: HR 1.21 (95% CI 1.00–1.47)		dichotomous ≤ 0.83; HR 1.36 (95% CI 1.23–1.50)
Kleczynski et al., 2017 [24]				linear: HR 2.83 (95% CI 2.01–3.98) dichotomous: HR 124.12 (95% CI 21.92–702.72)
Patel et al., 2019 [35]		OR 2.21 (95% CI 0.63–7.74)		
Sathananthan et al., 2019 [37]				OR 0.78 (95% CI 0.35–1.72)
A.J. Altisent et al., 2017 [28]	unable/slow/fast			slow walker: HR 2.30 (95% CI 1.35–3.93)
Green et al., 2013 [27]				unable: HR 1.85 (95% CI 1.26–2.72)
Alfredsson et al., 2016 [20]		per 0.2 m/s decrease: OR 1.16 (95% CI 1.06–1.28)		
Steinvil et al., 2018 [21]		dichotomous: OR 1.74 (95% CI 0.36–8.50)		dichotomous: OR 2.34 (95% CI 1.03–5.32)
Kano et al., 2017 [30]				dichotomous: OR 2.01 (95% CI 1.20–3.38)
van der Wulp et al., 2020 [22]	≤0.8 m/s			HR 2.5 (95% CI 1.4–4.5)
Green et al., 2015 [38]				per unit decrease: HR 1.37 (95% CI 0.53–3.45)
Green et al., 2012 [39]				each quartile: HR 1.19 (95% CI 0.82–1.66)
Assmann et al., 2016 [33]	0.75 m/s	HR 0.11 (95% CI 0.10–1.43)		
Saji et al., 2016 [36]	0.5 m/s		dichotomous: not predictive; *p* = 0.174	
Dvir et al., 2013 [29]	<50 m			<50m: HR 1.69 (95% CI 1.28–2.47)
Arnold te al. 2018 [31]	unable/quartile	linear: OR 0.95 (95% CI 0.89–1.02); dichotomous: OR 1.27 (95% CI 1.02–1.58)		
Shimura et al., 2017 [32]	n.s.	not predictive		
**Serum albumin**				
Bogdan et al., 2016 [42]	≤3.5 g/dL			baseline low level: HR 2.02 (95% CI 1.04–3.91)
Berkovitch et al., 2020 [43]				low level: HR 1.92 (95% CI 1.09–3.38)
Chauhan et al., 2016 [23]				HR 3.12 (95% CI 1.80–5.42)
Green et al., 2015 [38]				per unit decrease: HR 1.25 (95% CI 0.88–1.79)
Green et al., 2012 [39]				HR 1.51 (95% CI 1.03–2.21)
Kiani et al., 2020 [19]		dichotomous: HR 1.29 (95% CI 1.12–1.48)		dichotomous: HR 1.50 (95% CI 1.40–1.60)
Michel et al., 2019 [44]				dicohtomous: HR 2.10 (95% CI 1.53–2.87)
Patel et al., 2019 [35]		not predictive		
Shimura et al., 2018 [40]		HR 2.36 (95% CI 1.64–3.40)		not predictive
Steinvil et al., 2018 [21]		dichotomous: OR 8.21 (95% CI 1.04–64.70)		dichotomous: OR 2.21 (95% CI 1.12–4.37)
van der Wulp et al., 2020 [22]				HR 2.30 (95% CI 1.30–4.00)
Hermiller et al., 2016 [34]	≤3.3 g/dL	dichotomous: HR 1.60 (95% CI 1.04–2.47)		dichotomous: HR 1.40 (95% CI 1.04–1.91)
Forcillo et al., 2017 [18]	≤3.4 g/dL	per 1g/dL decrease: OR 0.26; *p* = 0.02		not predictive; OR 0.53; *p* = 0.07
Grossman et al., 2017 [41]	≤4.0 g/dL			per 0.5 g/dL decrease: HR 3.03 (95% CI 1.66–5.26)
Krishnan et al., 2019 [45]	n.s.			higher level: HR 0.30 (95% CI 0.20–0.50)
**ADL—Katz-index**				
Cockburn et al., 2015 [52]	score < 6/6	OR 1.07 (95% CI 0.64–1.77)		HR 0.86 (95% CI 0.71–1.05)
Dziewierz et al., 2017 [26]				dichotomous; HR 13.92 (95% CI 6.29–30.79)
Green et al., 2015 [38]				HR 1.59 (95% CI 0.93–2.70)
Green et al., 2012 [39]				HR: 2.13 (95% CI 0.97–4.71)
Kleczynski et al., 2017 [24]				per point decrease: HR 6.06 (95% CI 3.15–11.64) dichotomous: HR 20.06 (95% CI 6.93–58.02)
Kleczynski et al., 2018 [25]				predictive
Martin et al., 2017 [49]		per point drop: OR 1.27 (95% CI 1.11–1.44)		
Martin et al., 2018 [48]		OR 2.10 (95% CI 1.39–3.15)	HR 1.74 (95% CI 1.19–2.55)	HR 1.23 (95% CI 0.86–1.75)
Puls et al., 2014 [47]		HR 3.05 (95% CI 1.40–5.70)		higher score: HR 2.50 (95% CI 1.60–3.90)
Forcillo et al., 2017 [18]	score ≤ 4/6	not predictive		OR 0.80; *p* = 0.04
Steinvil et al., 2018 [21]		OR 2.43 (95% CI 0.58–10.20)		OR 1.43 (95% CI 0.59–3.45)
Goudzwaard et al., 2018 [51]		not predictive		linear: HR 1.50 (95% CI 1.21–1.90) dichotomous: HR 1.80 (95% CI 0.85–3.70)
Bureau et al., 2017 [50]	score < 5/6		predictive	not predictive
Chauhan et al., 2016 [23]				HR 2.45 (95% CI 1.42–4.22)
Szekely et al., 2019 [53]	n.s.			not predictive
**Handgrip strength**				
Chauhan et al., 2016 [23]	BMI/sex			HR 3.31 (95% CI 1.01–10.85)
Forcillo et al., 2017 [18]		not predictive		predictive
Green et al., 2015 [38]				HR 1.02 (95% CI 0.99–1.05)
Green et al., 2012 [39]				HR 1.18 (95% CI 0.84–1.66)
Goudzwaard et al., 2018 [51]		not predictive		
Hermiller et al., 2016 [34]		not predictive		not predictive
Steinvil et al., 2018 [21]		dichotomous: OR 2.24 (95% CI 0.28–17.80)		dichotomous: OR 1.63 (95% CI 0.66–4.06)
Dziewierz et al., 2017 [26]	weak/mild/strong			HR 28.84 (95% CI 10.54–78.87)
Kleczynski et al., 2017 [24]				HR 37.93 (95% CI 10.63–135.35)
Kleczynski et al., 2018 [25]				predictive
Shimura et al., 2017 [32]	n.s.	not predictive		
**IADL—Lawton index**				
Assmann et al., 2016 [33]	score < 8/8	HR 0.92 (95% CI 0.71–1.20)		
Boureau et al., 2017 [54]			not predictive	
Bureau et al., 2017 [50]			predictive (*p* = 0.065)	predictive (*p* = 0.0061)
Eichler et al., 2017 [12]				not predictive
Goudzwaard et al., 2018 [51]		not predictive		linear: HR 1.20 (95% CI 1.07–1.33) dichotomous: HR 2.30 (95% CI 1.06–4.90)
Schoenenberger et al., 2018 [55]				HR 1.23 (95% CI 0.67–2.28)
Schoenenberger et al., 2012 [14]			linear: OR 1.46 (95% CI 1.13–1.89) dichotom: OR 2.19 (95% CI 0.91–5.27)	
Stortecky et al., 2012 [9]		linear: OR 1.39 (95% CI 0.91–2.11) dichotom: OR 1.19 (95% CI 0.27–5.31)		linear: OR 1.25 (95% CI 0.92–1.70) dichotomous: OR 1.52 (95% CI 0.92–9.83)
van der Wulp et al., 2020 [22]				HR 1.50 (95% CI 0.90–2.30)
**Mini-Mental State Examination (MMSE)**				
Assmann et al., 2016 [33]	score < 27/30	HR 0.98 (95% CI 0.77–1.25)		
Boureau et al., 2017 [54]			HR 1.02 (95% CI 0.82–1.26)	
Eichler et al., 2017 [12]				not predictive
Goudzwaard et al., 2018 [51]		not predictive		dichotomous: HR 1.60 (95% CI 0.76–3.22)
Schoenenberger et al., 2018 [55]				dichotomous: HR 2.35 (95% CI 1.33–4.14)
Schoenenberger et al., 2012 [14]			linear: OR 2.64 (95% CI 1.55–4.50) dichotomous: OR 3.18 (95% CI 1.38–7.29)	
Shimura et al., 2018 [40]				not predictive
Stortecky et al., 2012 [9]		linear: OR 2.85 (95% CI 1.35–6.17) dichotomous: OR 7.62 (95% CI 1.44–40.19)		linear: OR 2.72 (95% CI 1.40–5.31) dichotomous: OR 2.98 (95% CI 1.07–8.31)
van der Wulp et al., 2020 [22]				not predictive
**Psoas muscle area index (PMAi)**				
Mamane et al., 2015 [58]	tertile/sex			female: HR 0.88 (95% CI 0.78–0.99)
Kleczynski et al., 2018 [25]				predictive
Kofler et al., 2018 [56]		L3: OR 0.082 (95% CI 0.011–0.589) L4: OR 0.049 (95% CI 0.005–0.536)		L3: OR 0.200 (95% CI 0.083–0.482) L4: OR 0.083 (95% CI 0.029–0.235)
Saji et al., 2016 [36]			HR 1.53 (95% CI 1.06–2.21)	
van Mourik et al., 2018 [57]		HR 0.32 (95% CI 0.05–1.91)		female mid-PMA: HR 0.14 (95% CI 0.05–0.45) female high PMA: HR 0.38 (95% CI 0.16–0.99)
Garg et al., 2016 [60]	2 groups (cut-off/sex)			not predictive
Krishnan et al., 2019 [45]				HR 2.50 (95% CI 1.10–4.60)
Foldyna et al., 2018 [59]	quartile/sex			HR 1.90 (95% CI 1.35–2.68)
Michel et al., 2019 [44]				not predictive
**Timed Up-and-Go test (TUG)**				
Boureau et al., 2017 [54]	≥20 sec		dichotomous: OR 0.39 (95%CI 0.11–1.41)	
Eichler et al., 2017 [12]				dichotomous: OR 5.12 (95% CI 1.64–16.01)
Goudzwaard et al., 2018 [51]		not predictive		linear: HR 1.10 (95% CI 1.02–1.09) dichotomous: HR 1.80 (95% CI 0.77–4.18)
Schoenenberger et al., 2018 [55]				dichotomous: HR 3.41 (95% CI 1.95–5.97)
Schoenenberger et al., 2012 [14]			linear: OR 1.64 (95% CI 1.26–2.12) dichotom: OR 4.23 (95% CI 1.83–9.77)	
Stortecky et al., 2012 [9]		linear: OR 1.83 (95% CI 1.10–3.05) dichotomous: OR 13.77 (95% CI 1.62–111.01)		linear: OR 1.74 (95% CI 1.24–2.45) dichotomous: OR 6.65 (95% CI 2.15–20.52)
Assmann et al., 2016 [33]	≤12.5 sec	HR 1.04 (95% CI 0.94-1.16)		
**Clinical Frailty Scale (CFS)**				
Miura et al., 2017 [63]	score ≥ 4/9			HR 1.84 (95% CI 0.45–7.55)
Yokoyama et al., 2019 [64]				HR 1.03 (95% CI 0.60–1.86)
Afilalo et al., 2017 [13]	score ≥ 5/9	dichotomous: OR 1.87 (95% CI 0.99–3.53)		dichotomous: OR 2.40 (95% CI 1.63–3.52)
Sathananthan et al., 2019 [37]				not predictive
Seiffert et al., 2014 [62]	score ≥ 6/9			per SD increase: HR 1.31 (95% CI 1.13–1.52)
Shimura et al., 2017 [32]	5 classes	HR 1.42 (95% CI 1.04–1.95)		per class increment: HR 1.28 (95% CI 1.10–1.49)
Honda et al., 2019 [61]	n.s.			HR 1.44 (95% CI 1.04–1.99)
Shimura et al., 2018 [40]				per point increase: HR 1.17 (95% CI 1.01–1.35)
**Canadian Study of Health Aging scale (CSHA)**				
Martin et al., 2018 [48]	score ≥ 5/7	OR 1.46 (95% CI 0.96–2.23)	HR 1.37 (95% CI 0.94–2.01)	HR 1.61 (95% CI 1.14–2.29)
Dziewierz et al., 2017 [26]				score > 5: HR 39.10 (95% CI 15.85–96.46)
Kleczynski et al., 2018 [25]	4 classes			predictive
Kleczynski et al., 2017 [24]				per point increase: HR 3.82 (95% CI 2.46–5.94) dichotomous: HR 64.65 (95% CI 17.35–240.94)
Cockburn et al., 2015 [52]	n.s.	OR 0.99 (95% CI 0.63–1.57)		not predictive
Martin et al., 2017 [49]		not predictive		
**4 items composite index**				
Afilalo et al., 2017 [13]	score ≥ 3/4	dichotomous: OR 2.65 (95% CI 1.28–5.49)		dichotomous: OR 3.04 (95% CI 1.98–4.66)
Huded et al., 2016 [68]		not predictive		
Okoh et al., 2017 [66]		higher score: HR 1.65 (95% CI 1.01–2.66)		score 4/4: HR 1.84 (95% CI 1.06–3.17)
Okoh et al., 2019 [67]				HR 1.84 (95% CI 1.23–2.69)
Green et al., 2015 [38]	score ≥ 5–6/12			linear: HR 1.12 (95% CI 1.02–1.22) dichotomous: HR 2.18 (95% CI 1.27–3.75)
Green et al., 2012 [39]		OR 2.20 (95% CI 1.02–4.60)		linear: HR 1.15 (95% CI 1.02–1.30) dichotomous: predictive tertile: HR 1.71 (95% CI 1.01–2.89)
Chauhan et al., 2016 [23]	n.s.	not predictive		score ≤ 2: HR 2.00 (95% CI 0.85–4.71) score 3: HR 3.05 (95% CI 1.24–7.46) score 4: HR 8.56 (95% CI 3.38–21.67)
**5 items composite index**				
Afilalo et al., 2017 [13]	score ≥ 3/5	dichotomous: OR 1.45 (95% CI 0.77–2.72)		dichotomous: OR 1.63 (95% CI 1.12–2.37)
Ewe et al., 2010 [71]				HR 4.20 (95% CI 2.00–8.84)
Rogers et al., 2018 [69]		dichotomous: OR 5.06 (95% CI 1.36–18.80)		dichotomous: OR 2.75 (95% CI 1.55–4.87)
Shi et al., 2018 [70]			OR 2.20 (95% CI 0.20–8.00)	
Steinvil et al., 2018 [21]		higher score: OR 15.30 (95% CI 2.71–86.10)		score ≥3: OR 2.23 (95% CI 1.14–4.34)
Abramowitz et al., 2016 [72]	n.s.			HR 2.04 (95% CI 1.31–3.20)
**Elderly Frailty Toolset (EFT)**				
Afilalo et al., 2017 [13]	score ≥ 3/5	dichotomous: OR 3.29 (95% CI 1.73–6.26)		dichotomous: OR 3.72 (95% CI 2.54–5.45)
Drudi et al., 2018 [73]		dichotomous: OR 3.50 (95% CI 1.74–7.07)		dichotomous: OR 3.33 (95% CI 2.21–5.04)
Pighi et al., 2019 [74]		per point increase: OR 1.27 (95% CI 1.07–1.50)		dichotomous: OR 1.83 (95% CI 1.33–2.50)
Sathananthan et al., 2019 [37]				per point increase: OR 1.72 (95% CI 1.39–2.14)
Skaar et al., 2018 [75]				HR 1.36 (95% CI 0.87–2.21)
**Bern scale**				
Afilalo et al., 2017 [13]	score ≥ 3/7	dichotomous: OR 3.29 (95% CI 1.53–7.07)		dichotomous: OR 2.57 (95% CI 1.69–3.91)
Eichler et al., 2017 [12]				not predictive
Schoenenberger et al., 2018 [55]				per IQR increase of 3 points: HR 3.29 (95% CI 1.98–3.91)
Schoenenberger et al., 2012 [14]			linear: OR 1.73 (95% CI 1.36–2.20) dichotomous: OR 1.69 (95% CI 1.32–2.16)	
Stortecky et al., 2012 [9]		per point increase; OR 2.18 (95% CI 1.32–3.61) dichotomous: OR 8.33 (95% CI 0.99–70.48)		per point increase: OR 1.80 (95% CI 1.31–2.47) dichotomous: OR 3.68 (95% CI 1.21–11.19)

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
