# Peer review of "The Pandora’s Box of Frailty Assessments: Which Is the Best for Clinical Purposes in TAVI Patients? A Critical Review"

_jcm, 2021, doi:10.3390/jcm10194506_

Round 1

Reviewer 1 Report

In the updated version of the manuscript the authors have enriched the paper with appropriate methodological considerations and a more detailed discussion of their findings. However, I still think there are unsolved critical issues. The absence of a quantitative statistical analysis represents the main methodological limitation of the study, which provides only limited information for the clinician. In this regard, the authors state that “the use of single frailty markers - also due to their good practicability - in clinical studies seems justified. For clinical decision-making, composite indices are more appropriate for the characterization of a patient”; however, I think that it is not correct to separate clinical research and clinical decision-making: conversely, the goal of clinical research should be to guide clinical decision-making in daily practice. Indeed, given the high complexity of frailty definition, the use of single variables, despite acceptable for clinical studies, does not sufficiently help the clinician in the assessment of frail patients, and even less in the challenging balance between utility and futility of percutaneous structural interventions.

Author Response

We understand the approach of the reviewer. Indeed, the non-feasibility of meta-analyses limits the significance of the study for the clinician. Please note, that this results from the enormous heterogeneity of studies found by systematic literature search as meta-analyses were planned part of study protocol. To outline the limitation, we state in the discussion: “However, due to the remarkable heterogeneity of the included studies, meta-analyses of the study results were not indicated. Therefore, this review does not provide pooled effect sizes for the frailty assessments investigated, potentially limiting the information for the clinician.” (line 356-359)

We totally agree, that the goal of clinical research should be to guide clinical decision-making in daily practice – as you wrote. However, research is small-scale and study objectives often do not match the objectives in clinical practice. Studies enrolled in our review investigated the predictive value of frailty (parameters) for mortality. In daily practice, not mortality but treatment for frailty or its components prior to TAVI will be in the foreground. To underline your intention, that single parameters cannot present the complexity of frailty, we extended our statement that you cited above this way: “Given the high complexity of frailty syndrome and frailty definition, for clinical decision-making, composite indices are more appropriate for the characterization of a patient. The use of single parameters, despite acceptable for clinical studies, is not recommended in this context.” (line 385-388).

Reviewer 2 Report

The authors investigated the frailty indices used throughout clinical studies in patients treated with TAVI. A total of 79 studies investigating frailty with 49 assessments were included. Single markers of frailty found predictive of mortality included gait speed and serum albumin. Besides, composite indices were also predictive of both 30-day and 1-year mortality.

Main comments:

The review is well written and demonstrates the importance of assessing frailty in patients who require TAVI.

I have minor issues:

Data presented in the tables are fully repited in the text. I would try not to replicate the full information from the table in the text to make it more friendly for the reader. A summary of most important results from each section would be fine in my opinion. 

Author Response

To the "main comments":

Thank you very much!

To the "minor issues":

Yes, we agree. We have shortened the text in the results section removing some redundant information.

This manuscript is a resubmission of an earlier submission. The following is a list of the peer review reports and author responses from that submission.

Round 1

Reviewer 1 Report

Title:The Pandora’s Box of frailty assessments: Which is the best for 2 clinical purposes in TAVI patients? A critical review

In the presented manuscript, the authors evaluated which frailty assessment tools are mainly investigated and of significant predictive value for short- and mid-term outcome in patients undergoing TAVI. The authors included 79 studies in their review and present interesting data, showing that frailty was mainly assessed using single markers, such as gait speed and levels of serum albumin, that predicted 1-year mortality, while composite frailty tools have been shown to be associated with short- and mid-term outcome after TAVI. However, the the present work is more of a summary than a review, as meta-analyses of the included study results could not be performed.

Although the presented data are interesting, there are some important issues, which the authors may like to address.

Concerns:

  1. In my view one of the most important finding is the marked heterogeneity in definition of frailty and the used cutoff values of the assessment tools. In the end, this meant that the authors were unable to perform meta-analyses. However, the authors should focus more on this point and try to better elaborate the reasons for this heterogeneity.

  1. The authors should better differentiate between frailty and malnutrition. Although frailty and malnutrition are connected syndromes with significant overlap in their pathophysiological pathways, they describe two different syndromes of the elderly.

  1. The authors observed thatsingle markers are the most frequently used tools to assess frailty and that they are associated with 1-year mortality in patients undergoing TAVI, which is absolutely correct. However, frailty is a multidimensional syndrome among older adults characterized by a marked vulnerability and diminished capability to recover from stressors. In this respect, I doubt the predictive value of single markers. This aspect should be discussed more critically.

  1. In daily practice, frailty and malnutrition are often not measured, due to the lack of consensus about the most suitable and practicable assessment tool. Which new aspects does the review provide in this regard? Can the authors make a recommendation based on their analyses?

  1. The study would benefit from a discussion of possible consequences of the findings. Should all patients undergoing TAVI be systematically screened for frailty and malnutrition beforehand? Is it possible to develop recommendations for therapeutic approaches to support these patients on the basis of the results of the study?

  1. The authors should add a recently published study by Al-Kassou et al on the topic, CCI 2021

Reviewer 2 Report

Baritello and colleagues in their manuscript entitled “The Pandora’s Box of frailty assessments: Which is the best for 2 clinical purposes in TAVI patients? A critical review” performed a systematic review of 79 studies on frailty assessment in old patients undergoing transcatheter aortic valve implantation (TAVI). Frailty was evaluated differently across studies and the different types of assessments were divided into three categories: single markers, multidimensional frailty scales, and composite indices. Authors found that frailty was most commonly assessed using single markers, especially with those evaluating physical activity (gait speed) and malnutrition. Specifically, slow gait speed and low pre-procedural levels of albumin were associated with a higher risk of 1-year mortality. Moreover, 5 composite indices were predictive of short- and long-term mortality.

The article is overall well written, and the topic of frailty assessment is of great interest in daily clinical practice. However, I think there are some critical issues to take into account before considering this paper for publication.

MAJOR COMMENTS

 The considerable heterogeneity between studies represents the strongest limit of the present paper. Indeed, the impossibility to perform a metanalysis downsizes the clinical meaning of the study, which provides only descriptive data and does not add substantial novelty to the current knowledge on frailty assessment.

Despite the high complexity of frailty definition (which goes far beyond the mere aggregate of the patients’ comorbidities), single markers appear to be the be the most frequently used tool to assess frailty in clinical practice: how do the authors explain this paradoxical finding? Can they further motivate the absence of solid multiparametric scores? Percutaneous structural interventions, albeit safe and effective, carry a high risk of short- and long-term complications, which should be considered, and weighted with the overall frailty of the patient, when evaluating the potential futility of the procedure, especially in older patients. Do the authors think that the adoption of single markers might be comprehensive enough to overcome the need for a global clinical evaluation of such high-risk patients? Considering the complexity of frailty definition, a systematic review might not be an adequate tool, while a patient-level metanalysis might be the correct methodological approach to deal with such a challenging topic and to provide novel insights for the clinician.

MINOR COMMENTS

 In the discussion section, authors state that “No randomized control studies were included in our review, and it is likely that patient selection in the observational cohort studies was already influ-334 enced by the decision of the heart team. This is inevitable, given the nature of TAVI as a treatment reserved for high-risk patients requiring valve replacement”. To date, the indication for TAVI is progressively increasing, considering that randomized data demonstrate the safety and efficacy of the procedure even in patients at moderate and low surgical risk. The authors are encouraged to update this concept in accordance with the recent evidence.

Please refer to TAVI as “transcatheter aortic valve implantation” and not “transcatheter aortic valve intervention”.